# Efficacy of Chondroprotective Food Supplements Based on Collagen Hydrolysate and Compounds Isolated from Marine Organisms [note 1]

**DOI:** 10.3390/md19100542

**Published:** 2021-09-26

**Authors:** Thomas Eckert, Mahena Jährling-Butkus, Helen Louton, Monika Burg-Roderfeld, Ruiyan Zhang, Ning Zhang, Karsten Hesse, Athanasios K. Petridis, Tibor Kožár, Jürgen Steinmeyer, Roland Schauer, Peter Engelhard, Anna Kozarova, John W. Hudson, Hans-Christian Siebert

**Affiliations:** 1RI-B-NT—Research Institute of Bioinformatics and Nanotechnology, Schauenburgerstr. 116, 24118 Kiel, Germany; thomasi-e@gmx.de (T.E.); zry147896@163.com (R.Z.); p.engelhard@kielnet.net (P.E.); 2Institut für Veterinärphysiolgie und Biochemie, Fachbereich Veterinärmedizin, Justus-Liebig-Universität Gießen, Frankfurter Str. 100, 35392 Gießen, Germany; mahenabutkus0@gmail.com (M.J.-B.); monika.burg-roderfeld@hs-fresenius.de (M.B.-R.); 3Department of Chemistry and Biology, University of Applied Sciences Fresenius, Limburger Str. 2, 65510 Idstein, Germany; 4RISCC—Research Institute for Scientific Computing and Consulting, Ludwig-Schunk-Str. 15, 35452 Heuchelheim, Germany; 5Tierarztpraxis Dr. Silke Fritscher, Bergstraße 104, 73441 Bopfingen, Germany; 6Animal Health and Animal Welfare, Faculty of Agricultural and Environmental Sciences, University of Rostock, Justus-von-Liebig-Weg 6b, 18059 Rostock, Germany; helen.louton@uni-rostock.de; 7Institute of BioPharmaceutical Research, Liaocheng University, Liaocheng 252059, China; 8Tierarztpraxis Dr. Karsten Hesse, Rathausstraße 16, 35460 Staufenberg, Germany; k.hesse@tiermedizin-drhesse.de; 9Medical School, Heinrich-Heine-Universität Düsseldorf, Universitätsstr. 1, 40225 Düsseldorf, Germany; opticdisc@aol.com; 10Center for Interdisciplinary Biosciences, Technology and Innovation Park, P. J. Šafárik University, Jesenná 5, 04001 Košice, Slovakia; tibor.kozar@upjs.sk; 11Laboratory for Experimental Orthopaedics, Department of Orthopaedics, Justus Liebig University Giessen, Paul-Meimberg-Str. 3, 35392 Giessen, Germany; Juergen.Steinmeyer@ortho.med.uni-giessen.de; 12Biochemisches Institut, Christian-Albrechts Universität Kiel, Olshausenstrasse 40, 24098 Kiel, Germany; e_schauer@t-online.de; 13Department of Biomedical Sciences, University of Windsor, Windsor, ON N9B 3P4, Canada; kozarova@uwindsor.ca (A.K.); jhudson@uwindsor.ca (J.W.H.)

**Keywords:** osteoarthritis, collagen hydrolysate, sulfated N-acetyl glucosamine, sialic acids, eicosapentaenoic acid (EPA), MMP-3, ADAMTS-5

## Abstract

Osteoarthritis belongs to the most common joint diseases in humans and animals and shows increased incidence in older patients. The bioactivities of collagen hydrolysates, sulfated glucosamine and a special fatty acid enriched dog-food were tested in a dog patient study of 52 dogs as potential therapeutic treatment options in early osteoarthritis. Biophysical, biochemical, cell biological and molecular modeling methods support that these well-defined substances may act as effective nutraceuticals. Importantly, the applied collagen hydrolysates as well as sulfated glucosamine residues from marine organisms were strongly supported by both an animal model and molecular modeling of intermolecular interactions. Molecular modeling of predicted interaction dynamics was evaluated for the receptor proteins MMP-3 and ADAMTS-5. These proteins play a prominent role in the maintenance of cartilage health as well as innate and adapted immunity. Nutraceutical data were generated in a veterinary clinical study focusing on mobility and agility. Specifically, key clinical parameter (MMP-3 and TIMP-1) were obtained from blood probes of German shepherd dogs with early osteoarthritis symptoms fed with collagen hydrolysates. Collagen hydrolysate, a chondroprotective food supplement was examined by high resolution NMR experiments. Molecular modeling simulations were used to further characterize the interaction potency of collagen fragments and glucosamines with protein receptor structures. Potential beneficial effects of collagen hydrolysates, sulfated glycans (i.e., sulfated glucosamine from crabs and mussels) and lipids, especially, eicosapentaenoic acid (extracted from fish oil) on biochemical and physiological processes are discussed here in the context of human and veterinary medicine.

## 1. Introduction

Osteoarthritis is the most common joint disease in humans and animals and shows increased incidence in older patients. Generally, NSAIDs (Non-Steroidal Anti-Inflammatory Drugs) are used in the treatment of osteoarthritis [1,2]. However, their use is problematic in that side effects cannot be excluded and liver and/or kidney damage can be present, particularly in geriatric patients [2]. Therefore, conventional painkillers are of concern and alternatives with less or no side effects are being sought. Chondroprotective compounds are considered to be well tolerated and can be administered unhesitatingly over longer periods of time. In this context, glucosamine sulfate as well as a special diet with fish oil belongs to a standard therapy in veterinary medicine. However, collagen hydrolysate has not been used in the same way of so-called chondroprotective drugs such as glucosamine-based nutraceuticals [3]. Nevertheless, collagen hydrolysates were analyzed on a submolecular level in various conditions [4,5,6,7,8,9,10]. Clinical studies in which potential effects of collagen hydrolysate and sulfated glucosamine are directly compared with each other and discussed in relation to molecular mechanisms are still missing. Beneficial effects of collagen- and proteoglycan-fragments may be related to specific interactions with receptors like integrins, especially in the case of collagen fragments [4] and aggrecan [11,12,13]. Beside these specific interactions, unspecific contacts between collagen-strands [14,15,16,17,18], proteoglycans and fatty acids within the extracellular matrix of the cartilage could also play a crucial role when explaining the therapeutic effects on a sub-molecular size level. Therefore, we assessed the effect of the applied substances on the cartilage health of animals under study by a combination of biophysical, biochemical, cell-biological methods and molecular modeling tools. Such an arsenal of methods has previously been used to assess the beneficial therapeutic effects of other bio-medical-relevant macromolecules. These macromolecules are hyaluronic acid, proteoglycans and phospholipid species [19], sulfated poly- and oligosaccharides [20,21,22,23], polysialic acid and sialic acid containg oligosaccharides [24,25,26,27,28] as well as lysozymes and anti-microbial peptides in complex with oligosaccharides [29,30,31]. We analyzed structure–function relationships and focused on molecular modeling calculations that would allow us to better understand the details of intermolecular interactions between sialic acids and relevant proteins. The collagen hydrolysate under study [32,33,34,35,36,37] was examined with high resolution NMR experiments, especially, DOSY NMR [4,7]. Additionally, molecular modeling approaches such as molecular docking and molecular dynamics simulations were carried out in order to obtain more information about the interaction potency of collagen fragments and sulfated GlcNAc with receptor structures. As a result of this study we are now able to formulate efficient encapsulation strategies [38,39,40] for an oral application of peptides and proteins [41,42,43,44]. The concept of this study was to provide a comparative examination of the potential benefits of nutraceuticals as chondroprotective agents. Thereby, we tested two specific collagen hydrolysates of bovine and fish origin, sulfated glucosamine from marine organisms as well as fish oil in lipid and vitamin enriched dog food. Collagen-hydrolysate is not a standard therapy in veterinary medicine for the treatment of osteoarthritis symptoms. We therefore carried out this study in both dogs and horses which both suffer from similar osteoarthritis symptoms to compare its efficacy to standard therapy (sulfated GlcNAc). Molecular modeling studies of the applied nutraceuticals are essential since they provide valuable hints on how these substances could influence biochemical processes not only in dogs and horses but as an extension of this study also in humans.

The aim of this work was to examine and compare the influence of collagen hydrolysate and sulfated glucosamine in dogs with osteoarthritis with regard to their ability to alleviate pain and to reduce the associated clinical orthopedic symptoms. In addition, a smaller control group (trailing group) was employed in order to avoid a placebo group. This group was administered a special food developed for joint protection since, due to ethical considerations, it was necessary that all dogs under study were treated for the symptoms of early osteoarthritis. The study was combined with cell assays and molecular modeling calculations with the information obtained, providing a foundation on how the results of this clinical study may be applied or be useful in the treatment of other species (e.g., horses) with the same or different collagen hydrolysates (e.g., from fish skin or from jellyfish collagen).

## 2. Results

### 2.1. Drug Administration and Sta†istical Analysis od Dog Treatment

All dogs (Table 1) were examined at the beginning of our study for the characteristic symptoms of osteoarthritis. The body condition score (BCS) of the dogs is defined according to Mele [45]. Level 1 (cachectic) to level 9 (obese). Level 5 is considered ideal.

Fifty-two dogs were treated during the 16 weeks of the therapy period. The treatment resulted in improvements in agility of animals found in all three groups. To assess pain, one of the symptoms of OA, and any therapeutic progress, palpation was performed independently for the left and right femoral joint throughout the study. Notably, a reduction in tenderness/pain was observed as early as four weeks into treatment (Figure 1). After 16 weeks (the end of the therapy), all groups (including the control group which contains only the half number of patients) exhibited a reduction in the sensitivity of their femoral joints to manipulation.

A key indicator of therapeutic value in any trial is the effect of the substances under study on the quality of life (QOL). We therefore assessed QOL using previously published guidelines [46,47,48,49]. The QOL score of dogs is shown on Figure 2. All substances under study (collagen hydrolysate, sulfated glucosamine as well as the special dog food enriched with fatty acids and vitamins) resulted in positive effects on the QOL score. Collagen hydrolysate led to the most promising results in terms of displaying moderate to minor symptoms or no joint problems at the focus of study.

Our observations suggest that the collagen hydrolysate applied in this dog-study contains bio-active fragments that have a beneficial effect on OA symptoms, similar to that observed for sulfated glucosamine. Cellular studies show that the collagen fragments in the hydrolysate are responsible for the effects within the extracellular matrix of the joint tissue. These effects can be supportive, non-supportive or even detrimental [8,9,10]. In order to establish the correlation between structure and function of bio-active components of the collagen hydrolysate applied in our study (Fortigel from Gelita) we further characterized the relationship by well-established protocols [4,7,8,9,10].

The grades, i.e., the degrees of lameness (DL) were determined by a veterinary expert. The orthopedic examination concerning joint pain symptoms (Figure 1) was carried out first. By presenting the dogs on a plane, non-slip floor covering at walk and trot, the degree of lameness was assessed. Both the orthopedic examination and the lameness assessment were documented on the standardized examination sheets by the veterinarian. The entire movement cycle (walk and trot) including walking upstairs was examined. Since dogs usually display a mixed lameness, it was not recorded whether this was a support-leg-lameness or a sloping-leg-lameness. The decisive factor is the degree of lameness, which is differentiated into five different degrees (zero means not detectable). The grade of lameness (DL) in dogs (Figure 3) with chronic musculoskeletal disorders is given on a scale from zero to four. Of the total of 52 dogs, 9 dogs showed no lameness during the entire study period (4 animals in the collagen group, 3 animals in the glucosamine group and 2 animals in the joint diet group). All other dogs were permanently or intermittently lame or had a stiff gait. DL 4 could not be determined in any of the test subjects. At week 0 (initial examination) a stiff gait could be observed in 3 dogs, DL 1 was present in 16 dogs and DL 2 was present in 22 patients. DL 3 was prevalent in two dogs. At week 16 (final examination), DL 3 was present in one dog, DL 2 in 15 animals, DL 1 in 11 animals and a stiff gait pattern in 2 dogs. A total of 19 dogs show no lameness (DL = 0).

Four dogs had to be removed from the study and were therefore not included in the final examination. No statistically significant differences were found in the distribution of the DL between the groups (*p*-value for week 0: 0.35; week 4: 0.85; week 8: 0.36; week 16: 0.59). There were roughly the same number of dogs free of lameness in all groups, with mild or moderate lameness. The lameness of the dogs decreased significantly during the investigation period (*p*-value 0.015), i.e., at the end of the investigation period significantly more animals were free of lameness than at the beginning of the investigation. The collagen and glucosamine groups had a comparable composition in regard to the degree of lameness. There were no statistically significant differences. Within the three groups, the distribution of the DL was shown as presented in Figure 3. In the group following the joint diet, only minor changes in the DL were found during the study period. During the final examination three animals were free of lameness, while at the beginning there were two in group 3. Within the other two groups, significantly more animals were free of lameness at week 16. In the younger dogs (2 to 5 years old), a relatively large number of them were still running free of lameness, despite the x-ray evidence of arthritis. With increased age of the animals, the number of dogs showing lameness increased. In particular, the proportion of dogs with DL 2 was frequently represented with advancing age. When comparing group 1 (collagen) with group 2 (glucosamine) at the end of the study period 9 in group 1 and 7 in group 2 are completely free from lameness.

The distribution of grades in regard to an assessment of complaints after invalidation/ burden is shown differentially by color at 4-week intervals during the sixteen-week study period (Appendix A). The complaints of all dogs under study after prolonged exposure were rated with an average mark of 2.92 at the beginning of the study period. In the collagen, glucosamine and joint diet groups the average mark was initially 2.64, 3.38 and 2.16, respectively. At the end of the study period, the average mark of all dogs was rated 2.42. In the individual collagen, glucosamine and joint diet groups, the average marks were 1.93, 2.8 and 2.46, respectively. As the diagram shows, the dogs in the glucosamine group tend to have more severe symptoms after prolonged exercise.

The distribution of jumping ability is shown differentially by color at 4-week intervals during the sixteen-week study period (Appendix A). The average mark for jumping was rated with a mark of 3.24 at the beginning of the study period. In the individual collagen, glucosamine and joint diet groups the average marks were 3.0, 3.63 and 2.86, respectively. At the end of the study period the average mark was 2.6, with the collagen, glucosamine and joint groups displaying marks 2.34, 2.79 and 2.68, respectively. Appendix A shows that the dogs in the glucosamine group tended to have more discomfort when jumping with a smaller number of animals jumping completely symptom-free.

The distribution of back pain sensitivity as revealed by the patient holders is shown differentially by color at 4-week intervals during the sixteen-week study period (Appendix A). The touch sensitivity of the back was rated with an average mark of 1.85 at the beginning of the study period. In the collagen, glucosamine and joint diet groups the averages were 1.89, 1.85 and 1.78, respectively. At the end of the study period, the average was 1.59 and characterized by a certain degree of homogeneity. At the same time the collagen, glucosamine and joint diet groups attained marks of 1.69, 1.48 and 1.63, respectively.

The distribution of joint pain sensitivity is shown differentially by color at 4-week intervals during the sixteen-week study period (Appendix A). The distribution of grades is shown differentially by color at 4-week intervals during the sixteen-week study period. The touch sensitivity of the affected joint was rated with an average mark of 2.19 at the beginning of the study period. In the collagen, glucosamine and joint diet groups the average was 2.63, 1.78 and 2.22, respectively. At the end of the study period the average was 1.97. In the collagen, glucosamine and joint diet the sensitivity score was 2.4, 1.63 and 1.94, respectively. By a descriptive point of view, a decrease in the sensitivity to touch was found in the collagen and glucosamine groups during the study. Overall, the difference in joint pain sensitivity between the groups was not striking at any time of the investigation (*p*-value 0.23 week 0 and 0.21 week 16). However, a possible trend is apparent with touch sensitivity decreasing for all three groups over the course of the study (grade 1.5–2.49).

The HCPI (Helsinki Chronic Pain Index [50]) of dog patients at the beginning and after 16 weeks of treatment is summarized in Table 2. We compared our data with the results of an article about the ameliorative effects of omega-3 concentrate in managing coxofemoral osteoarthritic pain in dogs [51]. Scores in the HCPI range from zero to four points, with four points corresponding to highest degree of pain. The points were averaged for each group with the expectation that with successful therapy the average score would decrease.

The impact of drugs and nutritional supplements on the mobility of randomly selected dogs of the study (Table 3) was evaluated and documented by video at both the beginning and end of the therapy. Our current observations are in full agreement with a horse-study recently published [35].

Representative videos (mp4 format) for two patients from the sulfated glucosamine group, two from the collagen hydrolysate group and one dog from the control group are available for download from the Appendix A.

Dog-patient handlers were able to easily perceive that dogs in the early stages of OA already exhibited difficulty with stair climbing. The videos clearly display this observation and demonstrate that sixteen weeks of therapy had a positive effect on the ability of these dogs to climb stairs along with no new outwardly discernable negative aspects or progression of disease.

### 2.2. X-ray and Statistics

X-ray data were recorded from each dog at the beginning of the study. There was no significant statistical deviation in the x-ray data when comparing the left with the right hip joint (Appendix A). This is also the case when comparing the left and the right knee joint (*p*-value left knee joint: 0.12; right knee joint: 0.13; left hip joint: 0.11; right hip joint: 0.15). We also recorded x-ray data from selected dogs at the end of the study. The number of dogs was limited due to the necessity of obtaining the permission of the patient holders. Furthermore, sedation of the dogs was necessary for X-ray imaging and our goal was always to reduce unecessary risk in the dogs under study. While the symptoms of pain and the QOL improved for a subset of the dogs under study, the X-ray data revealed that the damage to the bone in the affected joints was not improved with any of the treatments.

In relation to the X-ray analysis measurements of the left thigh circumference at the time of the initial examination resulted in an arithmetic mean of 39.7 cm (all patients combined). The smallest value was 24 cm, the largest 56 cm. The standard deviation was 8.1. A mean value of 38.5 cm was calculated for the collagen group, 39.6 cm for the glucosamine group and 42.5 cm for the joint diet group. At the final examination, the values were recorded in 46 patients. The mean value was 41.1 cm; the wingspan ranged from 21 cm to 57 cm. The mean value for the collagen group was 40.4 cm, for the glucosamine group it was also 40.4 cm and for the joint diet group it was 44.0 cm. The right thigh circumference at the time of the initial examination averaged 39.6 cm. In the collagen group it was 38.4 cm, in the glucosamine group it was 39.6 cm and in the joint diet group it was 42.2 cm. The smallest value was 24 cm, the largest 57 cm. During the final examination, the arithmetic mean of 40.98 cm was determined for all tested patients, 40.5 cm for the collagen group, 40.1 cm for the glucosamine group and 43.9 cm for the joint diet group. The wingspan ranged from 21 cm to 57 cm. The mean difference between the right and left thigh muscles at the initial examination, all groups combined, was 1.1 cm. The wingspan ranged from no difference up to 6 cm. In the collagen group the mean value was 1.3 cm, in the glucosamine group 0.9 cm, and in the joint diet group 1.1 cm. At the final examination, the arithmetic mean in the aggregation of all groups was 0.45 cm, in the collagen group 0.47 cm, in the glucosamine group 0.42 cm and in the joint diet group 0.5 cm. For the size of the left thigh muscles, the *p*-value for time was 0.0037 and the *p*-value for the groups was 0.67, i.e., there were only small differences between the groups that are not statistically significant. With regard to time, however, significant differences could be found, which means indicating a statistically significant increase in muscle size during the study period. The size of the right thigh muscles was similar. There were no statistically significant differences between the groups (*p*-value 0.68). Here, too, there was a significant increase in muscle size over time (*p*-value 0.01). The difference between the right and left thigh muscles (delta value) also shows no decisive differences in relation to the two main groups (*p*-value 0.33). Over time, however, there was a statistically significant reduction in the difference between right and left muscle circumference (*p*-value 0.0001).

X-ray data and measurements of the muscle circumferences provide physically measurable data, however, observation concerning pain symptoms and the mobility are also of high importance in the evaluation of the efficacy of nutraceuticals. The data were collected by the patient-holders (Figure 2 and Appendix A) but also by experienced veterinarians (Figure 1 and Figure 3) at the four examination times (initial examination, first examination—4 weeks, second examination—8 weeks, final examination—16 weeks).

### 2.3. Cell Biology Tests and Blood Parameters

The differentiation of canine as well as of equine chondrocytes was studied in the absence and in the presence of collagen hydrolysates and proteoglycan fragments. The distinct time-dependent differentiation pattern (e.g., with respect to the known sialic acid galactose linkage at the end of the saccharide chains of the corresponding glycoproteins is well established [24], and can be used to test the respective impact of various substances in cell culture very precisely [10]. A representative image displaying the expected pattern for equine chondrocytes grown on collagen is presented in Figure 4A.

The induction of multi-directional differentiation processes of equine and canine chondrocytes strongly depends on the kind of collagen hydrolysate [10], as shown in Figure 4A with the collagen hydrolysate under study. Furthermore, the nerve cell progenitor assay can act as a test system to control the sialic acid dependent impact of collagen fragments on cell migration and differentiation [24,26,52,53]. Figure 4B shows an example of a progenitor cell assay from cells of the subventricular zone of the mouse brain. This indicates that sialic acid staining is a feasible method to control the collagen dependent differentiation of glia.

Blood samples were analyzed for a homogenous group of dog patients (23 German Shepherd dogs) in which collagen hydrolysate was tested as a food supplement. Previous studies were focused on a correlation between MMP-3 plasma levels and MMP-3 synovia levels for dogs suffering from osteoarthritis [46,47]. Since MMP-3 is a highly proteolytic enzyme, enhanced breakdown of cartilage tissue in the German shepherd dog OA group could occur via degradation of collagen types II, IX, X [54] and aggrecan [55]. Additionally, TIMPs are known inhibitors of MMP within tissues. We therefore examined MMP-3 (Figure 5A) and TIMP-1 levels (Figure 5B) in a back-to-back study of 23 German Shepherd dogs (guard and protection dogs of the police) which were fed over a time of 8 weeks with the same collagen hydrolysate provided to the 20 dogs in the collagen hydrolysate group described above. The analysis of MMP-3 and TIMP-1 levels (presented in Figure 5) was performed in accordance with Parkkonen et al. [56]. Notably, MMP-3 levels were significantly reduced (*p* = 0.01) after 8 weeks of treatment. We did not find a significant alteration in TIMP-1 levels during this same period.

The data presented here also support the data gained earlier at Tierärztliche Hochschule Hannover [49]. Our results suggest that sulfated and non-sulfated glucosamines and small collagen fragments may have a direct influence on the activity of matrix metalloproteinases. Given this result, allosteric inhibition and stimulation as well as competitive inhibition were considered as an additional mechanism to be investigated when using collagen hydrolysates or proteoglycan fragments, i.e., sulfated and non-sulfated glucosamines as nutraceuticals.

### 2.4. NMR Analysis of Fortigel Collagen Hydrolysate

Our studies indicated that pain reducing effects are detectable by an evaluation of the mobility and agility of the animals under study. Furthermore, biochemical parameters are also altered as found in our analysis of the blood-probes with respect to osteoarthritis markers (e.g., TIMP-1, MMP-3). These observations are probably related to the occurrence of certain bio-active collagen fragments within the hydrolysates. It was therefore of interest to determine whether the positive effect on cartilage health for dog patients with beginning osteoarthritis symptoms can be correlated with the applied collagen fragment mixture. We therefore conducted TOCSY, NOESY and DOSY NMR experiments as described previously. Since the collagen hydrolysates differed in their composition, we further characterized them to provide a more detailed molecular analysis. In the present study, we used TOCSY and NOESY experiments to identify specific amino acid residues (e.g., Arg residues) of these bio-active compounds (Figure 6). This type of analysis allowed us to characterize the collagen hydrolysate unambiguously. The NMR results indicate that the size-range of the collagen fragments in the collagen hydrolysate food supplement is between 2.9 and 8.1 kDa, with no triple helical collagen structures present. Since the collagen hydrolysates differed in their composition, we further characterized them to provide a more detailed molecular analysis. In the present study, we used DOSY to identify specific amino acid residues (e.g., Arg residues) of these bio-active compounds (Figure 7). This type of analysis allowed us to characterize the collagen hydrolysate unambiguously.

### 2.5. Molecular Modeling

We utilized the available PDB structural data for MMP-3 (2JT6.pdb) and ADAMTS-5 (2RJQ.pdb) as the starting geometries for molecular modeling tasks. Although their experimental geometries also exhibit ligands in their binding sites, we were also interested in how sialic acid and GlcNAc in standard and sulfated forms can bind to MMP-3 and ADAMTS-5. Thus, we were equally interested in the prediction of all possible binding sites (BS) for these two proteins. Three binding sites were predicted using the SiteMap program for MMP-3, whereas the prediction of BS for ADAMTS-5 was three times higher. The number of predicted binding sites is in agreement with the size/weight of these proteins (18.54 kDa for MMP-3 and 42.84 kDa for ADAMTS-5).

In the next step, we calculated how N-Acetyl glucosamine (GlcNAc) and N-Acetyl neuraminic acid (Neu5Ac) (both in standard and sulfated forms) can bind into all predicted binding sites. We used the Glide program to determine binding poses and the energetics of binding for the four carbohydrates.

The Glide analysis for all binding sites and all carbohydrates resulted in more than one thousand protein–ligand complexes. Table 4 presents the lowest energy binding poses for MMP-3 and ADAMTS-5.

The ribbon presentation for MMP-3 together with the ligand–protein interaction analysis is illustrated in Figure 8A. Equivalent figures for ADAMTS-5 are presented in Figure 8B.

It is interesting to note Neu5Ac is predicted to bind preferably (apart from BS3 of ADAMTS-5) over GlcNAc. The sulfated forms of glucosamines in most cases bind better than the unsulfated molecules. The exception is Neu5Ac in BS1 of MMP-3. This is a special case (illustrated in Figure 8A-BS1) because this binding site appears below the protein loop and the carbohydrate molecule also interacts with the amino acids of the loop.

The next modeling step dealt with explicit modeling of the proteins with collagen fragments. We used the HEX program here to generate (based on shape and electrostatics complementarity) around 100 protein–collagen complexes for both proteins. The lowest-energy forms from the HEX modeling were used as the starting structures for molecular dynamics (MD) simulations.

MD simulations were performed in a water environment in order to evaluate the stability of the protein–collagen complexes. Figure 9 presents part of the results from a 50 ns simulation. The structure at the start of simulation plus 10 time-dependent structures extracted from the saved simulation trajectory at 5 ns intervals were superposed and are shown on Figure 9 in order to present the time-evolved conformational changes. The “simulation quality analysis” of Maestro/Desmond (Figure 8C and Figure 9A) indicated that the standard deviation of all analyzed parameters like total energy, potential energy or volume is below 0.01% of the average value variables that were a result of the MD simulations.

PLIP analysis and the consequent Access processing of the MD trajectory geometries allowed a comparison of overall hydrogen bonding versus hydrophobic interactions for both MMP-3/collagen and ADAMTS-5 collagen. Similar data were obtained with both systems with hydrogen bonding predominating with an incidence of 68% for ADAMTS-5 and 66% for MMP-3 complexes. In comparison, hydrophobic interaction stabilization accounted for 34% in the case of MMP-3 and 32% in the case of ADAMTS-5.

The carbohydrate entity present on the protein surface in the case of ADAMTS-5 interacts with the collagen structure as shown in Figure 8D and Figure 9B. Accordingly, the sulfate groups present at the glycan chains of proteoglycans can mediate interactions with collagen—triple helix structures of collagen present in cartilage.

## 3. Discussion

The ability to treat early stages of osteoarthritis with minimal side effects is a prime consideration in any treatment. Our study was focused on candidate dogs with early onset osteoarthritis.

Symptoms of pain noted during palpation by the attending veterinarian, mobility and agility marks evaluated by the patient-holders as well as an independent video-based assessment of the mobility of all the dogs in the study are displayed in Figure 1, Figure 2 and Figure 3. As revealed by the data in the glucosamine group and even more clearly in the collagen hydrolysate group: the lameness of dogs with mild symptoms clearly improved, Figure 3. Intermolecular interactions of the glucosamine entity present on the ADAMTS-5 protein surface (as presented in Figure 9B) can influence the triple-helical structure of collagen present in the cartilage. Beside the administration of glucosamine sulfate the interactions of collagen fragments with matrix metalloprotease–carbohydrate complexes underline the importantance of glycobiological aspects of cartilage health.

A visible difference in the thigh musculature between healthy and diseased legs clearly decreased in all three groups. Additionally, there was a striking reduction in pain during palpation of the knee joint in both the collagen group and glucosamine groups. During the study period, this was accompanied by a significant reduction in symptoms while standing, climbing stairs, jumping and after prolonged strain. An obvious reduction in touch sensitivity of the affected joint and the spinal cord, respectively, as well as an obvious increase in running pleasure was also found in these two groups. Similarly, a clear reduction in the symptoms while standing and a reduced touch sensitivity of the affected joint and an increase in running pleasure was found in the joint diet group. This group displayed minimal improvement in the other four parameters. The HCPI (Helsinki Chronic Pain Index; Table 2) and the QOL (Quality of Live score) improved in all three groups. The observed clinical improvements, particularly through the administration of glucosamine sulfate and collagen hydrolysate, indicate a positive effect for these two compounds.

It was impossible to detect the effect of the standard dog food in this study. In order to observe the effect of standard food a placebo group has to be created, comprising dogs fed in an ordinary way. This was forbidden due the fact that the animals were sick and needed to be treated. Consequently, group 3 was entitled as commercial joint diet group. It was the fatty acid-related group where we expected healing success as well. The food in group 3 was changed completely (no collagen or glucosamine enrichment) and the dogs were administered with power food only. The food for this group can be entitled as power food because it was designed to contain valuable fatty acids. The feeding in group 1 and 2 was different. The dogs obtained standard dog food but enriched with collagen hydrolysate (Group 1) or sulfated glucosamine (Group 2). While the feeding within Group 2 or Group 3 matched standard feeding protocols, fish-oil-enriched collagen hydrolysate feeding is more or less a novel protocol. Independently of the therapeutic options used for Groups 1 to 3, the beneficial effect of treatment (but slightly different) was monitored in all three groups of the dogs.

Our combined clinical, cell biological, biochemical, biophysical and molecular modeling approach on canine and equine patients is a feasible strategy to answer a number of questions related to collagen hydrolysates, sulfated glycans and lipids as chondroprotective food supplements. Articular cartilage destruction is mediated by the loss of collagen type II and proteoglycans and this loss is a characteristic feature of osteoarthritic (OA) symptoms. Our results show that it is possible to correlate the influence of collagen hydrolysates on cartilage tissue [32,33,34] through specific biochemical pathways and cell–biological processes [4,7,8]. We found that collagen hydrolysates were able to alter the levels of MMP-3 without changing the level of TIMP-1 (Tissue Inhibitors of MetalloProteinases-1; data not shown). In addition, we recognized the involvement of collagen hydrolysates and sulfated glucosamine in several key biochemical processes which are directly correlated with cartilage health. As noted, collagen hydrolysates contain mixtures of collagen fragments of various length. We show here that modeling is a useful tool in evaluating specific interactions in protein binding sites and interactions between these proteins and collagen fragments. The impact of sulfation in the case of glucosamine is discussed here in detail. In comparison to sulfated GlcNAc, we analyzed the interactions of sulfated Neu5Ac, which is an interesting member of the sialic acid family with bio-medical relevance [24,25,27,28,30,57]. In addition, the effect of chondroprotective collagen hydrolysate of fish or jellyfish origin on cell cultures is similar to the effect of collagen hydrolysate of bovine origin used in the present dog study. We note that the EU Community approval/registration of fish or jellyfish originated collagen hydrolysate is not completed yet. No doubt further investigation on the impact of these interactions on protein activity in relation to targeting diseases will be crucial in further evaluating their effect on the extracellular matrix and therapeutic value.

Cartilage markers (MMP-3 and TIMP-1, Figure 5A,B) were analyzed over a time period of 8 weeks from a homogenous group of 23 German Shephard dogs of nearly the same weight. This was important in order to determine a daily dose of 20 g. One could argue that a dose for smaller dogs with a lower body weight has to be adapted to these patients. Since collagen hydrolysate and sulfated glucosamine are nutrition supplements without any toxic effect, overdosing in smaller dogs theoretically could have led to increased positive effects. This, however, was not observed during our study. Cell assays, molecular modeling studies, examination of eight horses and one dog in a long-time study were necessary in order to find out in which way our results can be extended to different species (including humans) and to collagen hydrolysates from other resources (e.g., fish skin and jellyfish). Therefore, Figure 4A,B is not shown in order to document a state before and after a treatment with collagen hydrolysate. The effects of collagen hydrolysates on equine cells have been documented in detail in the literature (Raabe et al., 2010) [10]. It is also feasible to analyze the impact of collagen hydrolysate on the migration of nerve cells (Zhang et al., 2016) [26]. In this context Figure 4B demonstrates that it was possible to observe the differentiation of stem cells treated with collagen hydrolysate by staining with polysialic acid.

To figure out whether the positive effects of collagen hydrolysate may be detectable or useful in other species that display osteoarthritis related problems in their movements, we also treated selected horses (50 g/day for a normal-sized horse and 25 g/day for a smaller horse e.g., a Shetland pony) In the case of a Holstein horse, a Hanoverian horse, an Arabian horse, an American Quarter horse, a Trotter, an English Blood horse, a Shetland pony and a Trakehner horse positive responses were detectable with corresponding methods as discussed here for the dog patients.

Overall, all three therapies have a positive effect on dogs’ health, justifying their individual use. However, the data presented here along with the supplementary data indicated that sulfated glucosamine and collagen hydrolysate used as supplementary nutraceuticals are more effective than high-quality dog food alone, with collagen hydrolysate and sulfated glucosamine having similar positive effects on cartilage health. However, importantly the data indicate that supplementing a dog’s diet with collagen has the greatest effect on reducing lameness, thus suggesting that the goal of increasing collagen levels should be incorporated into a treatment regime that targets the OA symptom of lameness.

## 4. Materials and Methods

### 4.1. Dog Osteoarthritis and Drug Administration

Dogs were chosen based on a thorough anamnestic workup, a general and an orthopedic examination (including an X-ray examination) as well as a blood draw. Subsequently, 52 dogs were randomized into three groups. 20 dogs received collagen hydrolysate, 21 dogs received glucosamine sulfate and 11 dogs received the special diet. The animals in all three groups were fed with the supplements or the special diet over a period of 16 weeks. During this time, follow up examinations were carried out after 4, 8 and 16 weeks. Data from a separate study of 23 German Shepard dogs were used in our study to analyze the MMP-3 data over a time-period of 8 weeks. Furthermore, 8 horses (including a pony) were examined in order to determine the efficacy of this treatment design in other species.

Various dogs with early osteoarthritis (OA) were examined in a randomized clinical study examining the therapeutic effect of nutraceuticals. Specifically, the impact of (A) a collagen hydrolysate, (B) tablets of sulfated glucosamine and (C) a high-quality dog-food, i.e., Hills-JD (containing vitamins and enriched with fatty acids, especially EPA) on joint health was evaluated. The dogs were fed Hills-JD instead of a placebo, since the dogs already displayed osteoarthritis symptoms and were therefore already being treated accordingly. The main lipid-component of HillsJD is eicosapentaenoic acid (EPA) which is enriched in sea-fishes such as salmon and herring. The administered peptides, carbohydrates and lipids were delivered via the gastro-intestinal tract into the blood-stream and administered in this way to the crucial target tissues in the organism. A dose of 20 g collagen hydrolysate was administered per day and dog patient (based on the daily collagen requirements in food for wolves). The dose was not reduced for smaller dogs since a higher amount of this nutraceutical is completely harmless (as tested in a cell biological assays).

Sulfated GlcNAc in the form of tablets, collagen hydrolysate from Gelita dry powder, Hills JD high quality dog-food were used to target dogs’ osteoarthritis. Fish collagen hydrolysate is produced by the skin of deep water ocean fish (cod, haddock and pollock). The fish collagen (Norland Products Inc., Cranbury, NJ, USA) consisted primarily of alpha 1 and alpha 2 chains in a 2:1 ratio with a MW between 4.5 and 21 kDa [7,10].

The QOL-score (Quality of life) combines the mood of the animal, its mobility and agility, joy of playing, sounds of pain and problems with climbing stairs. Dog-patient handlers were provided with a questionnaire which evaluated the QOL score. In addition, to remove any perceived bias, a veterinary doctor, not involved in the study in any other capacity, assessed video footage of the treated animals to provide an independent QOL estimate.

### 4.2. Cell Biology Tests and Blood Parameters Determination

The cell biology tests used are the same as those described in our former publications Raabe et al. [10], Zhang et al. [26] and Petridis et al. [53]. We focused on the impact of collagen hydrolysates on the differentiation of chondrocytes as well as on the role of sialic acids as contact structures of the cell surface as differentiation markers.

Blood samples were obtained from all dogs and horses under study in order to clarify any existing medical condition. The blood samples were taken after stasis and disinfection with 70% alcohol on the anterior vein cephalic with a 7.5 mL S-Monovette (Sarstedt) and an attached cannula (Sarstedt). An amount of 16 IU served as anticoagulant Heparin. The blood was centrifuged at 3000 rpm for 10 min after collection. A large blood count (hematology), an organ profile and an IgG/IgM borreliosis antibody titer were created for each dog in the Synlab Augsburg laboratory. Blood was drawn as part of the treatment.

The samples obtained were labelled and stored in a deep freezer until evaluation.

MMP-3: A mouse anti-dog stromelysin-1 monoclonal antibody MAC-084 (UCB Celltech, Slough, UK) was used to coat the ELISA plates (Greiner Bio-One GmbH, Kremsmünster, Austria) by incubation of 10 g/50 mL PBS buffer. After overnight incubation at 4 °C., the blood plasma samples were applied undiluted to the ELISA plates and incubated at 4 °C for two hours. The second antibody, a rabbit anti-dog stromelysin-1 polyclonal antibody (Biotrend Chemikalien GmbH, Köln, Germany) was added at a 1:8000 dilution and incubated again for two hours at 4 °C. The last two-hour incubation at 4 °C was carried out with a peroxidase-labeled goat anti-rabbit antibody (Sigma Aldricks, Saint Louis, MO, USA) at a 1:20,000 dilution. The amount of bound peroxidase as a measure of the concentration of MMP-3 present in the sample was determined using tetramethylbenzidine as the peroxidase substrate. Three washing steps are carried out between each new antibody coating in order to separate unbound antigens from the sample. The enzyme reaction was stopped by the addition of sulfuric acid. Prostromelysin (UCB Celltech, Slough, UK) in a concentration of 126 ng/mL was used as the standard. The resulting yellow color change was measured at 450 nm in a Spectra Photometer (Tecan, Männedorf, Switzerland).

TIMP-1: The ELISA plates (Greiner) were treated with an anti-dog TIMP-1 monoclonal antibody MAC-080 (Celltech) in a concentration of 5.7 g/50 mL PBS buffer with incubation overnight at 4 °C. The blood plasma samples were applied to the ELISA plates at a 1:7 dilution. After a two-hour incubation at room temperature on the Thermostar (13MG), a rabbit anti-human TIMP-1 polyclonal antibody (Biotrend) in a 1:2000 dilution was used as the second antibody. After a further incubation of two hours at room temperature on the Thermostar, coating was carried out with a peroxidase-labeled goat anti-rabbit antibody (Sigma) at a 1:4000 dilution. The amount of bound peroxidase as a measure of the concentration of TIMP-1 present in the sample was determined using tetramethylbenzidine as the peroxidase substrate. Several washing steps were carried out between each new antibody coating in order to separate unbound antigens from the sample. The enzyme reaction was stopped by the addition of sulfuric acid. Recombinant dog TIMP-1 was used to generate a standard curve with a concentration range of 27.5 ng/mL. The resulting yellow color complex was measured at 450 nm in a Spectra Photometer (Tecan) [49].

### 4.3. Statistical Analysis

Preliminary data processing started with Microsoft Excel (Office 2000 package, Microsoft, Redmond, WA, USA). The biomedical data of animals under study were evaluated on the computers in the local computer network (LAN) of the Biomathematics and Data Processing unit of the Veterinary Medicine Department of the Justus Liebig University in Gießen. The statistical evaluations were carried out using the BMDP/Dynamic, Release 8.1 (Statistical Solutions Ltd., Cork, Ireland) [58] program package. Missing data in Tables are marked with *. Consequently, the given entry is treated as a missing value by the BMDP program. To describe the data, arithmetic means (x^−^), standard deviations (s), minima (x_min_), maxima (x_max_) and sample sizes (n) were calculated and presented in a table for quantitative, approximately normally distributed characteristics. The qualitative characteristics were counted separately according to groups and presented in the form of frequency tables. To statistically test the group and time influence for significance, a two-factor analysis of variance with repeated measurements with regard to time was carried out with the program BMDP2V in groups 1 and 2 with approximately normally distributed characteristics. If the values were missing, this was conducted using the BMDP5V (so-called “forest test”). With regard to the quantitative characteristics, the group comparison of these two groups with normal distribution used the t-test and otherwise the Wilcoxon-Mann–Whitney test (BMDP3D). For the semiquantitative characteristics, the exact Wilcoxon-Mann–Whitney test using the “StatXact” program (V1, Cytel, Waltham, MA, USA) was applied to compare the two groups. For the comparison of qualitative characteristics, frequency tables were generated for all three groups with the program BMDP4F. The qualitative characteristics were checked in the case of two expressions regarding significant correlations for each point in time only for groups 1 and 2 with the exact test by Fisher. The Fisher–Freeman–Halton test was used for more than two values. The “StatXact” program was used here (Cytel, 2010 [59]). The evaluation of the statistical significance was based on the significance level α = 0.05; this means results with *p* ≤ 0.05 were given as statistically significant. In addition, the exact *p*-value was given, if possible. Group 3, as a trailing group, was not included in the significance calculation.

### 4.4. Nuclear Magnetic Resonance (NMR) Spectroscopy

Proton NMR was applied to analyze the Fortigel collagen hydrolysate in terms of its fragment size distribution (DOSY [7]) and possible identification of certain amino acid types (TOCSY/NOESY). Mass-spectrometry and NMR previously provided a detailed molecular analysis of the collagen hydrolysates under study [4,7,8,9]. In the NMR tubes, collagen hydrolysates were dissolved at an amount of 3 mg in 0.5 mL water (90% H_2_O/10% D_2_O). The NMR experiments were performed on a 600 MHz Bruker Avance III spectrometer (Bruker, Karlsruhe, Germany) at 298 K. 2D-TOCSY experiments (DIPSI-2; mixing time 80 ms) and 2D-NOESY (mixing times 200 or 400 ms) were recorded with 512 (F1) × 1024 (F2) complex data points and a spectral widths of 7212 Hz (12 ppm). Water suppression was performed using excitation sculpting and, per increment, 16 scans were accumulated with an inter-scan recovery delay of 1.5 s. For processing we used zero-filling to 1024 (F1) × 2048 (F2) data points prior to Fourier transformation, followed by baseline correction in both dimensions. Spectra were calibrated on internal water.

### 4.5. Molecular Modeling

The structures of glucosamines, i.e., N-Acetyl Glucosamine (GlcNAc) and N-Acetyl Neuraminic acid (Neu5Ac; also sialic acid) were downloaded from PUBCHEM (https://pubchem.ncbi.nlm.nih.gov/, accessed on 12.07.2019) in SDF format and imported into the Maestro V. 12.3.013 (Schrödinger LLC, New York, NY, USA) [60] project table. Both molecules were then sulfated using the Maestro molecular builder option. Their molecular structures are presented in Figure 1. An advanced conformational search for carbohydrate side chain orientations were then carried out maintaining their ^4^C_1_ forms for GlcNAc and GlcNAc-sulf and ^1^C_4_ for Neu5Ac and Neu5Ac-sulf ring conformations. The geometries of five selected low-energy conformations of the four carbohydrates were ab initio (DFT B3LYP 6-31G**) optimized (releasing all geometric parameters) with the Gaussian [61] program (G09 version, Wallingford, CT, USA) and were then used as input ligands for molecular docking into matrix proteins like MMP-3 and aggrecanases (e.g., ADAMTS-5), present in the dog and horses organisms.

The atomic coordinates of MMP-3 (2JT6.pdb [13]), ADAMTS-5 (2RJQ.pdb [12]) and a collagen fragment (1EI8.pdb [62]) structures were downloaded from the protein database and imported into Maestro. All protein geometries were processed (adding missing atoms, fixing bond orders, assign partial charges) by “Protein Preparation Wizard” of the Maestro program (V12.3.013, Schrödinger LLC, New York, NY, USA) [60].

The possible binding sites of MMP-3 and ADAMTS-5 were evaluated using the SiteMap program (V3.9, Schrödinger LLC, New York, NY, USA) [63,64] of Schrödinger LLC. The G09-optimized conformations of four ligands (Figure 1) were docked into all SiteMap predicted binding sites of MMP-3 and ADAMTS-5 using the GLIDE program [65,66,67,68]. While flexible docking was considered for all side chains of the carbohydrates, the pyranose rings were fixed in 4C1 for 1C4 forms as stated above. All protein–ligand complexes resulting from GLIDE flexible were optimized (OPLS-2005 force field) and were tableted/ordered according their GLIDE docking scores.

The HEX program [69] was used to generate and preoptimize the geometries of the MMP-3/collagen and ADAMTS-5/collagen supramolecular complexes. From around 100 generated complexes the lowest-energy ones were selected for further molecular dynamics (MD) studies.

These HEX-output files were imported into Maestro and were processed by standard methods (as indicated above by “Protein Preparation Wizard”) to obtained the structures prepared for MD runs using the Desmond [60,70] program The OPLS-2005 force field (the up-to-date version of the OPLS force field family [71,72]) was used to carry out the simulation studies. The protein–collagen complexes were initially solvated in Maestro (SPC water model [73] with the water molecules added within a 1 nm buffer around the proteins) and the resulting structures were then minimized and equilibrated for 5ns. The final structures after equilibration were submitted for 50 ns NPT (pressure at 1.01325 bar) molecular dynamics (MD) simulations with the Desmond program at 300 K. Molecular geometries resulting from simulations were saved at 10 ps intervals and were used for further analysis. These geometries (5000 altogether) from each simulation were exported into pdb format and were used for analysis of interaction profiles using the PLIP program [74]. The xml files resulted from the PLIP calculation were imported into Microsoft Access for analysis and data mining.

Schrödinger’s Maestro [60] was used for visualization of molecular structures, their complexes and protein–carbohydrate ligand interaction profiles.

## 5. Conclusions

The experimental and computational methods applied were found to be useful tools in providing valuable information on the relationship between therapeutic value and mechanism. Such approaches will be key prior to these or other similar compounds being adopted further as potential medical therapies.

Collagen hydrolysate and sulfated glucosamine and potentially the fatty acid and vitamin rich food diet in the control group show similar benefits with respect to a treatment of early osteoarthritis symptoms. The positive effects in all three groups seems to be comparable, however, in contrast to sulfated glucosamine, collagen-hydrolysate provides more options for an improved therapy. Collagen hydrolysates are defined mixtures of short, long and medium-sized collagen fragments. Fortigel, used here, is a well-defined collagen hydrolysate that can be considered the gold-standard. Our NMR and molecular modeling techniques in combination with cell assays described here were essential in better defining possible mechanisms contributing to the improvement of the dog-patients. Furthermore, they provide a foundation for follow-up studies. These follow-up studies would include further examining the effect of this treatment in other species (e. g. horses) and a comparable analysis of various collagen hydrolysate compositions (e.g., fish skin, jellyfish). We note that even if the collagen-hydrolysates are from one species their compositions can be different depending on the formulation used by the supplier [8].

The positive effects we observed in the glucosamine (group 2) and the trailing group (group 3) were not surprising given their known use in the treatment of osteoarthritis. However, the clear and positive results in the collagen hydrolysate group (group 1) were not predictable at the beginning of the study. During the 16 weeks study period muscle and leg circumferences increased and pain symptoms were significantly reduced with this treatment. The study also clearly shows that there is variability among individual dogs, with some reacting well while others react weakly to the nutraceuticals under study. Since this may be based on the individual genetic makeup of the dog and subtle differences in the stage of disease having a third option becomes a desirable option that may provide added benefit to some animals. Additionally, sulfated glucosamine belongs to the standard therapies in the treatment of early osteoarthritic symptoms while only limited information exists about the molecular background of its beneficial effect, especially with respect to the impact of the sulfate group. Additionally, in the case of collagen hydrolysate, which is not part of a standard therapy, a number of structural questions which are related to its therapeutic functions are still open. Therefore, we combined the feeding study with intense molecular modeling calculations describing the stability of, e.g., complexes of MMP-3 with the nutraceuticals of group 1 and group 2. These data have provided us with an initial insight into unravelling key mechanisms of action for these compounds and avenues for future study and for treatment.

## Data Availability

The data presented in this study are available on request from the corresponding authors.

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
