# Peer review of "Efficacy of Chondroprotective Food Supplements Based on Collagen Hydrolysate and Compounds Isolated from Marine Organisms"

_marinedrugs, 2021, doi:10.3390/md19100542_

Round 1

Reviewer 1 Report

The authors did a good job in integrating most of the comments provided and now the paper is stronger. There are still, however, some major issues related to the statistical analysis. 

You report that you used Mann-Whitney U test, but MW test is only used when you have two independent populations. In your case, you have repeated measures on the same individuals (usually 4 measurements apart from 2 measurements represented in fig 5). You have either to use non-parametric tests for dependent samples (Friedman ANOVA for the samples for which you have more than 2 measurements, Wilcoxon paired test for the samples for which you have 2 measurements), or, better, Generalised Linear Mixed Models. Again, some statements that were unsupported before are still unsupported now (e.g. the ones relative to fig 1 - e.g. line 117- and fig 2 - statistics not present). I may sound pedantic, but having a correct statistical analysis is key to support your statements, and until you have correct analyses (I would prefer GLMM but I understand that can be complex) I cannot suggest the acceptance of your manuscript. 

Also, I am confused, relatively to fig 5, as usually box and whisker plots represent median, quartiles and range. Are you sure you represented means, SE, and SD? Also, if you use not parametric tests you should represent data via medians, quartiles, and range. You represent data via means and SD (or SE or 95% CI) when you represent normally distributed data (so if you show a t-test for example). You usually represent estimated model means and SE or 95% CI when you represent GLMMs. 

A tiny thing, relative to fig 6 and 7, you said you edited in the letter but you forgot to edit in the manuscript. 

Reviewer 2 Report

The authors did a lot of change and now the manuscript is better than the first version. The authors increased data, tables, description of data and the quality of the manuscript but some of them could be improved to publish it.

The introduction

This part was rewriter and now shows more useful information. But I think that could be interesting include the reason of they decided focus their present work in dogs and horse

Results

In line 111 “Overall, more than 80 dogs were treated during the 16 weeks therapy period with” but in line 388 “Subsequently, 52 dogs were randomized into three groups”, what it is real?

Lines 156-158 “To provide a less cluttered image the representation of the standard deviation (SD) 156 and the standard error (SEM) is omitted in the figures Fig. 1 - 3 and Fig. S1 - S4 but de- 157 scribed in the text or in the figure legends” but these data appear in supplementary material and in the text, authors describe the average mark but not the SD or SEM. Now the p values are included, I think that only the significance values should be included

Figure 6 in the legend are described A and B, but in the figure and in the text did not appear these letters.

In the results part the references are not necessaries only some of them as the corresponding to a software or something similar.

Discussion

Lines 384-395 include all the animal´s description and these paragraph are essential for the understanding of the manuscript, but I think that this information is not Discussion is Material and I recommend put all of them in Material as I describe in the next part.

Only the last part of the last paragraph should be consider Discussion

This part now is something better than in the first version but I think that the author continues to work on it. As for example move some data described in the result part In line 97 “Our current observations are in full agreement with a dog- and a horse-study recently published [32,33]” this type of data is more appropriated in discussion than in results part.

Material and Methods

Lines 449-476 now have more information and description than before. Although in a part before of Material they can described the N of dogs N of horses, and the numbers in each group, and the general characteristic of each animal of the animals that are included in each group.

I have the same question how do classify the early OA in dogs?

Line 482 “Blood samples were obtained from all dogs under study in order to clarify any existing medical condition and to ensure that there was no acute Lyme disease” and samples from horse?

Lines 493-494 described…”Cartilage markers were analyzed in accord of formerly published data [58-60]” this references described this methodology in synovial fluid to characterization of cartilage markers in blood or serum follow the same procedure?

Line 502 “Group 3 was not included in the significance calculation as a trailing control group.” The authors explain me….Due to ethical constraints a non-treated placebo group was not permited during the study since all the animals were suffering from joint pain and suffering must be treated. Therefore, the dogs in the control group received a high quality dog food which contained fish oil and vitamins. Regardless, when comparing this group to the groups that also received collagen and glucosamine several important insights were apparent with this information the Group 3 what type of dogs include? This is could be easier if the authors included a table with the groups and the name of the groups, because in the first part of Methods receive the name with letters and here with numbers.

Reviewer 3 Report

My major concern about this manuscript is the absence of statistical analysis. It is mandatory in a manuscript to perform appropriate statistical analysis. Moreover, it is also not clear, why the authors report that they treated horses and a pony but these data are not shown.

My comments are as follow:

Introduction

1) I suggest to use the word “osteoarthtritis” istead of “arthrosis” throughout the manuscript.

2)Lines 56-64: the authors should provide references.

3) The authors should decide if using “collagen hydrolysate” or “collagen-hydrolysate” and be consistent.

4) At the end of the introduction, the authors should report the aim/s of the study and not discuss the results. This part should be better organized and some points needs to be moved to the discussion.

Results

5) Line 106: the authors should publish the video as supplementary files. When I check the videos, the dropbox link was not working. 

6) Table 1: it is unclear why the authors reported Paula as an example. This dog (N=1) was treated with a mix of collagen-glucosamine-lipids. Thus, it should be deleted.  

7) The authors should add a full description of the animals used. Data should be added. Breed of dogs, age etc. Descriptive analysis is completely missing and needs to be added.

8) The authors enrolled animals affected by early OA. All the animals?

9) Figure 1: statistical analysis is completely missing. It should be performed and added. It is not acceptable to say there is an improvement regarding pain, without demonstrating the improvement by statistical analysis. I suggest to use a score to determine pain in animals ( for example, no pain=0; minor pain = 1, strong=2 and extra strong=3). Then, the authors could add a table reporting all the numbers (frequency and percentage of dogs for each grade of the scale and mean or median with SD or interquartile range according to the distribution of the data) and compared the data for each time point to see if there is or not a real improvement.  

10) I noticed that not all the dogs completed the follow-up. This should be specified.

11) Lines 137-141: this part should be moved to the methods.

12) Again, no statistical analysis was performed regarding QOL-score. The authors should better explain the score in the methods. Then, I suggest to add a table reporting the mean/median with SD or interquartile range for each time point and perform ANOVA or Kurskall wallis test with a post hoc analysis for multiple comparison to verify if there is a significant improvement.

13) No statistical analysis was performed regarding the degree of lameness.

14) Lines 189-191: “The number of dogs showing lameness increased with higher age of the animals. In particular, the proportion of dogs with DL 2 was correlated with advancing age.” The authors should add the analysis of correlation (rho and p-value are missing).

15) Lines 191-192: it is not clear to me how the authors evaluated these complaints. The methodology should be explained in the methods.

16) Line 200: how did the authors evaluated jumping ability?

17) Line 208: how did the authors back pain sensitivity?

18) Lines 191-226: the authors reported mean values of complaint, jumping ability and pain sensitivity. However, no statistical analysis has been reported. Again, the authors should compare the data using statistics.

19) The authors evaluated HCPI (Helsinki Chronic Pain Index) of dog patients at the beginning and after 16 weeks of treatment. Again, the authors reported the number but did not perform the analysis. I suggest to compare the delta (HCPI 16 weeks-HCPI baseline for each Behavior) between the different groups using ANOVA or Kruskall wallis test depending on the distribution of the data that should be checked by the authors.

20) Did the authors perform a x-ray of the dogs at the end of treatment? Did the authors analyze if there was a radiological worsening of osteoarthritis? 

21) The authors reported that they treated also horses and a pony. However, no data are reported. Thus, the authors could show the data (with analysis) or delete this part.

22) Section 2.2: The part on cells is unclear. In particular, it is not clear what the authors wanted to say and what they did. Did the authors studied the differentiation of canine as well as of equine chondrocytes in the presence of collagen-hydrolysates and proteoglycan fragments? In this case, the authors should report all the protocols used in the material and methods (cell isolation (stem cells?), cell cultures condition, cell diffentiation media, immunofluorescence protocol etc). Figure 4 is not clear. Why did the authors report an image of equine chondrocytes grown on collagen media? What is reported in figure 4b? The authors are supposed to report figures of the results obtained for this study. In figure 4, the authors are expected to report images of the cell before and after the differentiation and report experiments that support as collagen-hydrolysates and proteoglycan fragments impacts on cell differentiation (specific markers evaluation etc).

23) Lines 255-257: “Our cell biological studies demonstrated that the collagen-hydrolysate applied in this dog study had no toxic effects.”. It is not clear, where the authors demonstrated this point and how.

24) Line 270: the exact p-value should be reported and the statistical test used should be specified in the statistical analysis paragraph of the methods.

25) While it is clear the usefulness to study the composition of Fortigel collagen-hydrolysate, it is unclear why the authors reported molecular docking analysis in this study. The authors have a lot of data and a manuscript should be well-organized and homogenous with a logical thread. This study is focused on the effects of the diets on the dogs. Thus, the authors should be focused on this aim.

Discussion

26) The discussion should be modified accordingly to the statistical analysis that will be added. Moreover, a paragraph regarding the limitations of this study should be added.

Methods

27) In general, the authors should report all the necessary information to replicate the experiments.

28) In section 4.1 there is a lot of confusion. It should be better organized. First the authors should report all the necessary information about dogs. Age of the 52 dogs should be defined and reported. Breed of dogs should be specified. After the part related to the dogs, the authors should report the part related to the horses and pony. Moreover, the authors should add a paragraph regarding the outcomes that they analyzed (pain etc).

29) Lines 459-462: “We sought to conduct a comparative analysis of the potential benefits of these compounds and to determine the molecular interactions between collagen-fragments, parts of the glycan chains and lipids in the extracellular matrix, in order to provide novel information about the biochemical mechanisms which underlie the maintenance of mammalian cartilage tissue.” This part should be deleted. The authors should report the methods.

30) Line 483: cell biology tests should be briefly reported, even if the authors cited other papers.

31) Lines 495-498: “A portion of this blood was portioned for the determination of the cartilage markers and shipped to Gießen, where the cartilage analysis was completed in the laboratory of the Biochemical Institute of the Justus-Liebig-University.”. The authors should report what cartilage markers were analyzed and the kits used.

32) Section 4.3. Data Analysis, should be reported at the end of the methods and should be titled as “Statistical Analysis”. This part needs a lot of improvement. It is not acceptable to report only “The statistical evaluations were carried out using the BMDP / Dynamic, Release 8.1 [59] program package.”. The authors should describe in details the types of statistical tests applied and when.

Other comments:

Line 67: “submollecular” should be corrected.

line 452: “includung” should be corrected.

Line 476: A full-stop is missing.

Line 481: “4,5” should be corrected to “4.5”.

Line 481: references are not clear.

Line 511: “H2O/ 10% D2O” should be corrected.

Reference 46 is in German. I suggest to report only studies published in English.

Round 2

Reviewer 1 Report

The issues with the statistical analyses are unfortunately still there. It is difficult for me to give suggestions as the authors seem to miss the concept of repeated measurement. In figure 1-3 you show data on 52 dogs at three different experimental conditions (independent) BUT you collect data on each dog at four different time periods (0/4/8/16) even if you seem to have missed some measurements in the last week as the N is a bit different. This data collected on the same individuals at different times is called repeated measurement.

If you want to include statements such as “After 16 weeks (the end of the therapy), all groups (including the control group which contains only the half number of patients) exhibited a significant reduction in the sensitivity of their femoral joints to manipulation.” at lines 124-126, you need to consider that your data are repeated measurements on the same individuals and you need to test the effect of time periods.  

You report in your rebuttal letter that “Figures 1-3 are based on the personal evaluation of animal handlers. As such we do not think that higher level of analysis than SD values is appropriate for his scenario.” Surely there is a level of bias from a subjective measurement (not much if the assessment is done by the same person), but you can still test statistical differences, otherwise you cannot conclude that all treatment groups had a significant reduction. Reporting SD is descriptive statistics, not inferential statistics, so is not enough to infer statistical significance (95% CI can be considered enough).

Since you have 2 or 3 missing dogs, the best approach would be a generalised linear mixed model that can deal with missing values. For example, if we consider fig 1, your dependent variable is ordinal (a score from 1 to 4), you need to prepare the dataset this way (just showing dataset heading):

ID   |  TREATMENT CONDITION  |  Week  |   Pain level (1-4) RIGHT JOINT  |  Pain level LEFT JOINT

So, for each individual you have a score in each week and you have the associated dependent and independent variables. You need to use a mixed approach with individual as random effect.

The Mann-Whitney U test is only used to compare two independent distributions (not three), while you have three treatment conditions and four repeated measurements for each condition. Even if you want to compare the scores between time 0 and time 16 (that are not two independent groups), you will need at least a non-parametric test for repeated measurement (the simplest correct test that you can use with your data), not a Mann-Whitney U test. Furthermore, you will need to correct for multiple hypotheses testing (e.g., Bonferroni-Holm correction) as you test the effect of time in three treatment groups. That is why the mixed model approach is the most suitable in your case.

In Fig. 5 you changed mean with median, but that confuses me more. A box plot as I said before is representing medians, quartiles, and range. It is very weird that you represent median, SE, and SD.

Reviewer 3 Report

The manuscript was not improved enough. There are still problems with statistical analysis. The reading is difficult and the study is not well-focused. 

My comments are as follow:

Paragraph 2.1 should be rewritten in order to be clear. For example, the authors first describe pain symptoms of the dogs (lines 116-123 of the PDF) and then they describe how pain was assessed at lines 157-160.

Even if in the revision I asked to the authors to perform statistical analysis. No statistical analysis was performed regarding pain symptoms and quality of life of the dogs reported in Figure 1 and Figure 2. Did the authors report the p-values of pain symptoms at line 225 instead of reporting this data where the authors describe pain (lines 116-123)?

Lines 173-174: “19 dogs had an DL of 0. 4 dogs had to be removed from the study and were therefore not  included in the final examination.” This sentence is not clear.

Regarding the degree of lameness, the authors reported “The lameness of the dogs decreased significantly during the investigation period (p-value 0.015), i.e. at the end of the investigation period significantly more animals were free of lameness than at the beginning of the investigation.” . This is too vague.  What did the authors compare? Is there a difference between the groups? This is not clear to me. The authors reported “The collagen and glucosamine groups had a comparable composition with regard to the degree of lameness. There were no statistically significant differences.”. Therefore, it is not clear if there is a difference or not regarding lameness in relation to the different diets.

The authors analyzed complaints (lines 190-214) reporting averages. However, no statistical analysis comparing the averages between the three groups on the different time points was performed to establish if diet has an impact on these variables.

In the revision, I asked to the authors to perform statistical analysis on HCPI. Again, no statistical analysis was performed. The authors reply “Statistical considerations: Marks given by the patient holders which could be considered as a limitation of the study due to the possibility of subjectivity.”. Yes, I agree with the authors that marks were given by the dog holders and this is a limitation. However, statistical analysis must be performed. Subjectivity as a limitation should be specified in the limitation section of the discussion.

The section of cell biology tests is not useful. The authors should describe new results and not simple report two images. As the authors stated, the impact of hydrolyzed fish collagen on the differentiation of induced of equine adipose tissue-derived stromal cells in chondrocytes was already investigated by the same authors in 2010. Thus, this part should be deleted.

Lines 314-316: “Our cell biological studies demonstrated that the collagen hydrolysate applied in this dog study had no toxic effects.” The authors stated in the reply to reviewer that they deleted this sentence considering that they did not report these data.

The part on molecular docking analysis is not useful for this study and create confusion in the reader. A manuscript should be readable, fluent, well-organized and focused. I suggest to delete it.

In the methods, there is no mention about how the authors measured MMP-3 and TIMP-1. Did the authors use Elisa kits? This should be reported. It is not sufficient to write that they sent the samples by post.

Round 3

Reviewer 1 Report

I redirect the editor to made a choice on this paper, I cannot review further this paper. I believe the authors should consult someone with a good knowledge of statistics to help with the statistical analysis. Just avoiding terms such as statistical significance and removing information on the statistical tests used does not help if you still claim to have reductions/increases as a reader would expect them to be significant. The statistical approach is not transparent, and must be improved. The authors now present the same data without clear indications on the statistical tests used, so the paper is not improved, it is just less transparent. You now have several p-values in the results but you do not specify which tests you used. I still think the paper can be a good addiction to the literature but the issue with the statistical analysis is evident and the authors did not attempt to solve it (descriptive statistics is not enough to support your claims). I believe, by looking at the data, that most of the claims done by the authors cannot be supported by statistics (e.g., the reduction in the sensitivity in the femoral joints shown in figure 1 is not really statistically significant by looking at the data, while you claim that there is a reduction). 

Author Response

Dear Reviewer 1

Please note that Dr. Klaus Failing, the statistic expert of Vet. Med. Faculty of the Justus-Liebig-University of Gießen, Germany. has personally edited the statistics-related paragraph in our paper. Thank you for your guidance which has enabled us to improve our manuscript.

With kind regards

This manuscript is a resubmission of an earlier submission. The following is a list of the peer review reports and author responses from that submission.

Round 1

Reviewer 1 Report

n the present study the authors described the interest in evaluate the role of animal food enriched and how different fragments of these supplement could interad with molecule tring to find an explanation of the data described.

The manuscript ,in my opinion, need to revise the methodology, results and discussion and all of then could be to improve. Following with the data described by the authors in Line 343-344 the title of the manuscript could be revised.

1. Introduction

This part described the state of the art, althought are divided in two part and the authors include to aims of the work, I think that could be better described only one.

I have a question because I never work with dogs, How to evaluate the early OA in dogs? How to classify the dogs in the different groups of treatment?

2. Results

This first part of the Results 2.1. Drug Administration and Dog Studies (from Line 119 to 142), in my opinion could be the first part of Material and Methods. The authors could be describe the number of samples used in this work, characteristic of the population and the experimental desing, in this sense I think that some data are missing as for example, the total number of animals included in the study (Line 158 Overall, more than 80 dogs were treated during the 16 weeks therapy period with improvements in agility found in all three groups), the concentration of nutrients used and the description of each group.

In Line 159 the authors mentioned 3 groups but in the Table 1 appeared 4, the data and organization of the work are some confused. Line 163 “...all groups (including the control group which contains only the half number of patients)” but in the figure 1 does not appear the control group. For these, I think that it is necessary a table with the description of groups with the different traetments analized during the work.

The second part of results, 2.2. Cell Biological Tests. Started with “The differentiation of canine as well as of equine chondrocytes were studied in the absence and in the presence of collagen-hydrolysates and proteoglycan fragments” but in no where of the document the authors decribed these protocol of isolation, concentration of collagen hydrolysta and/or proteoglican fragmentes concentration or time used. In the figure 3 panel A, the authors showed a representative of equine chondrocytes and in panel B stem cells….I did not understand the relevance of this figure, why did the authors decide used here chondrocytes from horses and stem cells? In this part described that they inducted multi-directional differentiation but again data and protocol did not appear. In this block the results obtained did not describe

2.3. Blood Parameters In this case the number of dog analyzed are included. In the text are described that the measure of MMP and TIMP were developed in different times, but only appear data description at 8 weeks after treatment. The authors should include the Figure 4 in the text.

The figure 4 need more data in the description, for example. are arithmetic mean and SD represented in the graph?. In the blood sample only these two parameter were analyzed, in the case of yes the methodology corresponded to this part should be re-witter.

2.4. NMR Analysis of Fortigel Collagen-hydrolysate. Line 250 “collagen structures present [1]” why appear here this reference?

Line 250-251 “We note that mobility and agility are not automatically related to cartilage health” the authors said this, but how did they arrive to this conclusion?

Lines 251-252 “They might depend on a placebo effect or a general pain reduction as well as physiological improvements” some part of this block (as this line for example) could be better in the discussion than in results.

Figure 6 has two panels but he footprint only reflect this description “Parts of a two-dimensional NOESY spectrum of the used collagen-hydrolysate, Fortigel® . F1 and F2 are the 265 frequency axis which display the chemical shifts of the collagen protons (in ppm)”

Line 257 “This type of analysis has allowed to characterize the collagen-hy- 257 drolysate unambiguously” I have not experience in this type of analysis, could the authors explain me how arrive to this affirmation.

2.5. Molecular Modeling.

Line 268-269 “We utilized the available PDB structural data for MMP-3 (2JT6.pdb) and ADAMTS- 5 (2RJQ.pdb)” Why did the authors decide analyze here ADAMTS-5 and no in the rest of the article?

3. Discussion

The discussion is too short, the authors could move some parts of the results to here

Line 351 “We found that collagen-hydrolysates were able to alter the levels of MMP-3 (as well as Matrix Metallo-Proteinase-1 and -13; Figure 4) “. The data corresponding to MMP-1 and 13 did not appear in the manuscript. I think that in the discussion is not necessary include the reference of the figures.

4. Materials and Methods

4.1. Cell Biological Test

Line 365-366 “The methods used are the same as described in our former publications Raabe et al. [8] and Zhang et al. [24] “. In these publications used Adipose-derived stromal cells (ADSCs) and neuronal cells respectively, the authors in this manuscrit described, in the results, the data obtained from chondrocytes under differentiation. I my opinion the methodology for this experiment should be included.

4.2. Blood Parameter Determination.

Line 383 Cartilage markers were analyzed as published [53], but this reference (Taams, L.S.; Vukmanovic-Stejic, M.; Smith, J.; Dunne, P.J.; Fletcher, J.M.; Plunkett, F.J.; Ebeling, S.B.; Lombardi, G.; Rustin, M.H.; Bijlsma, J.W., et al. Antigen-specific T cell suppression by human CD4+CD25+ regulatory T cells) do not describe the cartilage markers analyzed, the authors could be revised the reference or described the methodology

4.3. Statistical Analysis

The Statistical Analysis methodology is in line 384 in stead of at the end of the manuscript.

Line 409 “Group 3 was not included in the significance calculation as a trailing control group.” The author could be described the groups with details, why did not use the control groups?

Reviewer 2 Report

The authors present some interesting data but the paper overall is very difficult to follow as there are no research questions, the introduction is not well structured, the discussion is very limited, and some of the results are not supported by statistics. For these reasons, I suggest not to consider the paper in this current form. I also think that the paper needs a lot of work that cannot be done in 10 days (the discussion is very short while the data presented are many). Also, some statements are not supported by statistical tests, so the paper needs to be re-evaluated. I thus suggest rejection with the possibility of resubmission. 

Specific comments:

  • Very long title, suggest to reduce to two lines.
  • The abstract should have a better structure with an introduction, research questions, methods, results, discussion. Now not clearly structured, missing introduction and discussion. 
  • Lines 60 and 64. Missing references.
  • I am unsure why you put the description of what you are presenting in the paper at the end of the first paragraph. Usually you start with a broader introduction and then going more in depth in your questions and what you want to describe later. The research question(s) and predictions are usually the last paragraph.  I suggest changing the structure.  
  • Line 75. Again, first you introduce your topic and make the bases for your questions/predictions, and at the end of the introduction you state your aims and justify your predictions. Please restructure your introduction.
  • Line 86. You again present your aim in the middle of your introduction. I suggest having just the final paragraph with what you are going to present, the importance of your study, and your predictions.
  • In general, you alternate statements supported by 10 references to unsupported statements. You need to be more balanced with your citations.
  • Line 137. Where are the assays? You should also include more information about the ethics. You have an ethics statement but no further information is provided. Also need ethics clearance number from the University.
  • Lines 137-140. Where are all these data? 
  • Line 148. Usually the links to data are in supplementary materials or data availability statement. 
  • Line 164. You need to support your statement with statistics. In figure 1 you just show the percentages of dogs with a certain score. But you do not present any statistics to support your claims so cannot conclude that there is a significant reduction.
  • Line 184. Again unsupported statements. Without statistics you cannot support your claims. You just present percentages. How do you test if they are statistically different?
  • Line 228. Which test used? Depending on the test please add the related important information. More details are needed.
  • Line 229. Why not shown? Which test used?
  • Figure 4. What are the values? Medians, quartiles, ranges? Which test did you use to compare them? 
  • Figure 5. What is "our dog study"? Please be more specific.
  • Figures 6 and 7. Please add more information, figures should stand alone.
  • Line 339. Why is it important? Which questions can be addressed? Why are they important? Did other studies highlight the importance of those questions? 
  • Line 341. Why? Can you please discuss and put into a broader literature context?
  • Line 346. Why do you assume that? Why is it important?
  • Line 349. Here you say your results but cite other papers. To which results do you refer to?
  • Line 352. Here you need to specify why it is important. You just summarise your findings without describing why they are important and without putting them into a wider context. Why can other researchers extrapolate from your study? You should specify your research questions and predictions in the introduction so that you can make a more coherent discussion.
  • Line 355. Here again, why is it important? What did other studies find? 
  • You need to discuss all your findings, not just part of it. Now completely missing the discussion on section 2.1 of your results (that is not supported by statistics). Also, the discussion on the other results is very weak.
  • From your concluding paragraph it seems you are presenting a method paper. This should be just a discussion point adding the following: How your tool is useful? Which applications?
  • Regarding the statistical analysis section in the methods, please update it when you make the new tests. Also, some details are not necessary like the use of Excel for data preparation, or that you marked missing data with *. I am not sure why you also add details that are not corresponding to the results. For example, where is the result of the MWU test? For which variables did you calculate descriptive stats (mean and sd, and range), not clear. Please check that you actually did what you put in there.